# Preparation and Hardened Performance of Bentonite-Induced Porous Magnesium Oxysulfate Cement Paste

**DOI:** 10.3390/ma15196872

**Published:** 2022-10-03

**Authors:** Tianyuan Xu, Jun Jiang, Guanghua Xiang, Jingchi Li, Zhongyuan Lu, Jun Li, Tao Ding, Luo Lei

**Affiliations:** 1State Key Laboratory of Environment-Friendly Energy Materials, Southwest University of Science and Technology, Mianyang 621010, China; 2School of Materials and Chemistry, Southwest University of Science and Technology, Mianyang 621010, China

**Keywords:** bentonite, thermal conductivity, mechanical strength, porous magnesium oxysulfate cement paste, pore structure

## Abstract

Porous magnesium oxysulfate (MOS) cement pastes were successfully fabricated by injecting presaturated bentonite into modified MOS cement paste. Their pore structure and hardened performance were investigated. The results indicated that the 20MgO-MgSO_4_·7H_2_O-18H_2_O system modified by citric acid (C_6_H_8_O_7_⋅H_2_O) and ethylene diamine tetraacetic acid was suitable to fabricate porous MOS cement paste. Bentonite slurry led to significant refinement of pores, generating nanosized pores in MOS cement pastes. When volume replacement of bentonite slurry in MOS cement paste rose between 0 and 60%, pore size corresponding to the peak in the pore size distribution curve of MOS cement-based materials decreased from 180.0 nm to 22.8 nm and then increased to 163.0 nm, and the porosity linearly increased from 21.1% to 58.1%. These small pores caused the successful preparation of porous MOS cement paste with dry bulk density of 760–1650 kg/m^3^, compressive strength of 7.8–69.8 MPa, and thermal conductivity of 0.25–0.85 W/(m·K).

## 1. Introduction

Cement-based porous materials have been extensively developed and applied in building external wall or roof thermal insulation during recent years, due to their fire resistance, safety, simple process, and low cost [1,2]. However, some critical problems associated with these thermal insulation materials still remain, such as a significant emission of greenhouse gases and high energy consumption in Portland cement production [3]. Thus, new types of cementitious materials with lower temperature production and emission of greenhouse gases are attracting much attention from scientists and environmentalists. Magnesium oxysulfate cement (MOS) as a new type of air-dried cement is still produced using an aqueous solution of MgSO_4_ and active MgO [4,5]. Although the emitted CO_2_ in the active MgO production process is higher than that in the Portland cement production process, because of 1 ton CO_2_/ton Portland cement and 1.7 tons CO_2_/ton MgO [6,7,8] with the help of carbonation of MOS cement and low temperature calcination, CO_2_ emission was reduced to at least 40–50%, and thus lower than that of Portland cement [6,7,8]. In addition, MOS cement materials also show excellent hardened performance, good cohesiveness, excellent electrical insulation, good fire and water resistance, lightness, and low thermal conductivity [9,10]. Furthermore, compared to magnesium oxychloride cement, MOS cement is generally unable to return chlorine, and has no corrosive effect on reinforced bars (steel bars) [11]. Due to these peculiarities, MOS cement has been highly studied in recent years, and thus scholars have pointed out that MOS cement can be used to produce porous materials to replace Portland cement-based porous materials in the thermal insulation field. 

Currently, many methods can be used to prepare porous MOS cement-based materials, for example, (I) substitution of lightweight fillers for normal-weight aggregate or MOS cement paste—this type is often treated as lightweight MOS cement paste [12]; (II) introducing prefabricated foam or forming air bubbles in MOS cement matrix—this type of lightweight material is often called foamed MOS cement paste or aerated MOS cement paste [13]; (III) eliminating fine aggregates from the MOS cement-based concrete—this material is often known as no-fines MOS concrete, but it cannot be used as a thermal insulation material due to the large content of interconnected air voids [14]. Introducing hydrogen peroxide or injecting prefabricated foam into the MOS cement matrix can form air bubbles to produce porous MOS cement-based materials [15]. A thermal conductivity of 0.12–0.89 W/(m·K), dry bulk density of 450–1685 kg/m^3^ and compressive strength of 1.1–47.4 MPa can be obtained by this method mentioned above [16]. However, air bubbles in the MOS system are often unstable, and bleeding, coalescence, and disproportionation often occur, leading to increased difficulties in controlling the air void structure [12,17,18]. For lightweight MOS cement paste, the substitution of air bubbles by lightweight fillers (such as rice husk and hemp hurd) can be used to solve problems caused by unstable air bubbles. Barbieri et al. and Qin et al. used bioaggregates as fillers to fabricate lightweight MOS cement-based materials with thermal conductivity of 0.11–1.06 W/(m·K), compressive strength of 0.5–70.4 MPa, and dry bulk density of 433–1721 kg/m^3^ [19,20]. However, these methods all build large voids in the MOS cement matrix, leading to a large reduction in strength, because voids are often seen as harmful to strength. Additionally, coarse voids always lead to large thermal conductivity, which is detrimental to the improvement of thermal insulation performance [12,13,14,15,21,22]. To further improve thermal insulation and mechanical performance, seeking approaches for preparing lightweight MOS cement paste with large content of small pores has become vital. 

Usually, introducing tiny pore-forming agents into matrix is one of the simplest ways to construct tiny pores, and Jiang et al. and Lu et al. show that bentonite slurry and aerogel may be emerging as tiny pore-forming agents for tiny pore construction [14,21]. This is mainly attributed to aerogel containing a high content of nanopore structures, so injecting aerogel into the MOS cement matrix means introducing nanopores into the MOS system. Different from aerogel, presaturated bentonite contains many montmorillonite nanolayers and excess water; this water cannot be completely consumed by cement hydration reaction, and thus generates a high percentage of big capillary pores. Nanolayers existed in these big pores may subdivide and refine these big voids to construct microsize or nanosize pores; importantly, it is cheaper than aerogel, thus showing greater potential in the construction of tiny pores in the MOS system to further enhance hardened performance [21,23]. Additionally, pores formed by pore-forming agents are different from traditional approaches; they can avoid using air bubbles to form pore structures, thus showing great advantages for porous MOS cement paste fabrication [21,24].

In the paper, MOS cement paste was firstly prepared by adjusting mix design parameters, then presaturated bentonite was introduced into MOS cement paste to construct tiny pores and to prepare porous MOS cement paste. Its pore structure, thermal insulation, and mechanical performance are investigated. We expect that the results in this research will present a new approach for the fabrication of porous MOS cement paste with excellent hardened performance, contributing to building thermal insulation.

## 2. Material and Methods

### 2.1. Materials and Preparation

Commercial light-burnt MgO (LBM) was purchased from Yingkou, Liaoning Province, China. Its particle size distribution and mineral phase are presented in Figure 1 and Figure 2. The chemical composition of LBM is presented in Table.1. In this study, magnesium sulfate (MgSO_4_⋅7H_2_O) is analytically pure and was purchased by Kelong Chemical Reagent Co., Ltd., Chengdu, China. Citric acid (C_6_H_8_O_7_⋅H_2_O) and ethylene diamine tetraacetic acid were used together as modifiers to improve MOS cement performance, which were all provided by Kelong company, mentioned above. Bentonite was provided by Weifang Shengshi Co., Ltd., Weifang, China (Figure 1). Its chemical compositions and particle size distributions are presented in Figure 2 and Table 1.

Porous MOS cement pastes were designed using a volumetric method in this study; their mix parameters are shown in Table 2. Fresh porous MOS cement slurry consisted of bentonite slurry and MOS cement paste. Prior to the experiment, cement matrix was firstly optimized by changing mix proportion parameters, such as MgO to MgSO_4_ ratio (M/S), H_2_O to MgSO_4_ ratio (H/S), and composite modifier (citric acid: ethylene diamine tetraacetic acid = 1:1), as shown in Table 2. Then, presaturated bentonite (Table 2) was manufactured by adding bentonite-to-water, stirring for 1 h, and then placing in undisturbed mode for 24 h for bentonite hydration. MOS cement paste was manufactured at 20 ± 5 °C; magnesium sulfate and modifier were first added to water to obtain MgSO_4_ solution and then added to a blender. Subsequently, LBM was added into the mixer, and stirred with tap water until an even MOS paste was fabricated. For porous MOS cement pastes, various compositions of bentonite slurry (Table 2) were immediately introduced into the MOS cement paste and stirred for about 60 s; subsequently, these pastes were poured into 40 × 40 × 160 mm^3^ molds and cured at 60 ± 10% relative humidity (RH) and 20 ± 2 °C environment for 24 h. Afterward, these samples were taken out the molds and cured in same environment (20 ± 2 °C, 60 ± 10% RH).

### 2.2. Test Procedures

The hardened performance of the samples was evaluated with mechanical strength tests after 3 days’ and 28 days’ curing, in accordance with ISO 679:1989 [25]. Three samples of each mix proportion and preparation condition were measured with DRE 2C equipment using transient mode (Xiangtan Xiangyi Instrument Co., Ltd., Xiangtan, China) to obtain thermal conductivity values (following with ISO 22007) [26].

Raw materials for this study were dried in 45 °C environment for about several days, until the mass did not change. X-ray fluorescence spectrometer (XRF) produced by PANalytical (Amsterdam, The Netherlands) was used to determine chemical compositions of these raw materials. LBM and bentonite particle were dispersed into anhydrous ethyl ethanol. Then, the particle size distribution was obtained with Mastersizer 3000 (Malvern Instruments Co., Ltd., Malvern, UK) under ultrasonic dispersion mode.

Cement hydration in samples was firstly terminated by immersing anhydrous alcohol, and then the sample was dried at 45 °C to the constant mass. Mineral phases of bentonite, LBM, or porous MOS cement paste were characterized using PANalytical X’PertPro (PANALYTICAL, Eindhoven, Netherlands) diffractometer (Cu target, step size: 0.03°). For the X-ray diffraction test, the bentonite sample was firstly crushed and sieved to take about 5 g of the sample. Then, the powder was placed in a test tube and filled with distilled water to make a clay suspension with ultrasound dispersion for 10 min, and then the suspension was centrifuged to separate out particles larger than 2 microns and smaller than 2 microns. Finally, oriented slices were used to put these particles on to slowly heat (45 °C) until mass was unchanged, then they were cooled and removed for testing. Data analysis was carried out using Jade software. For other samples, they were dried under 45 °C to constant weight and tested and analyzed using the above methods.

Microstructures of bentonite and the samples were observed and imaged via scanning electron microscopy (SEM) performed with a MAIA3LMU microscope (Tescan, Brno, Czech Republic). The conditions for SEM measurements were selected in vacuum mode, with an accelerating voltage of 15 kV and a working distance of 10 ± 2 mm. Samples were prepared without polishing.

In addition, pore structure of hardened samples was tested using AutoPore IV 9500 mercury intrusion porosimetry (MIP, applied intrusion pressure:0–30,000 psi) fabricated by Micromeritics (Norcross, GA, USA). Pores are usually treated as ideal cylindrical tubes, and the intrusion pressure (*P*) is associated with pore size (*d*) via the Laplace equation:(1)$$P=\frac{4\gamma \cos\theta }{d}$$
where γ is the surface tension of Hg, and θ is equal to 140° and is the contact angle of imperfect wetting between pore internal surface and Hg. Pore size distribution (PSD) of samples can be expressed as the dependence of the intrusion volume on the pore size. 

## 3. Results and Discussion

### 3.1. Optimization of MOS Cement Matrix

Mixture proportions influence cement hydration and microstructure of MOS cement paste, further affecting its hardened performance, which may influence the preparation of preparation of porous MOS cement pastes with excellent hardened performance. Therefore, suitable mixture parameters should be first identified to obtain matrix with good hardened performance. In addition, strength was simultaneously used to optimize the mixture proportions of the MOS cement pastes because of potential applications in the future.

#### 3.1.1. MgO to MgSO_4_ Ratio

Active MgO is highly related to hydration phase formation, but it cannot be accurately measured, so various MgO contents should be used to prepared MOS cement matrix for obtaining high-strength cement matrix. Usually, the M/S ratio is used to reflect the variation content of MgO, which is one of the most important mix parameters for MOS cement pastes. As shown in Figure 3, increasing the M/S ratio from 15 to 20 leads to the increased compressive strength of MOS cement matrix at the same H/S ratio and curing ages. Specifically, the 28-day strength of paste varied from 21.1 MPa to 28.4 MPa (H/S = 12), 20.6 MPa to 28.3 MPa (H/S = 15), 23.3 MPa to 39.6 MPa (H/S = 18), and 18.5 MPa to 22.3 MPa (H/S = 20), respectively. For 3 days’ curing, it shows the same tendency, and the strength respectively changes from 10.1 MPa to 17.4 MPa (H/S = 12), 10.1 MPa to 17.7 MPa (H/S = 15), 12.3 MPa to 22.1 MPa (H/S = 18), and 7.0 MPa to 10.7 MPa (H/S = 20). Further increasing the M/S ratio to 25 is adverse for the compressive strength, and the 3-day and 28-day compressive strength at the same H/S shows decreased trend (Figure 3). This is mainly attributed to the changes in content of active MgO. Because active MgO highly affects the formation of hydration phases, such as 517 phase, 318 phase, 115 phase, or Mg(OH)_2_, a low M/S ratio means a low content of active MgO, influencing the amount of hydration products, but a high M/S ratio indicates that more active MgO was introduced into the magnesium sulfate solution, and the excess active MgO caused a significantly increase in Mg(OH)_2_. However, Mg(OH)_2_ is porous and loose, which is often regarded as the low-strength hydration phase in the MgO-MgSO_4_-H_2_O system, leading to the decreased strength of the samples. Combining the results above, the suitable M/S ratio is 20.

#### 3.1.2. H_2_O to MgSO_4_ Ratio

The M/S ratio produces a significant influence on the compressive strength. Moreover, the H/S ratio play a vital role in hydration reaction controlling, because it decides the concentration of MgSO_4_, which determines the migration speed of the virous ions. Figure 4 shows the influence of H/S ratio on the compressive strength. The compressive strength of cement paste increased with the increase in H/S ratio from 12 to 18, but more water in system generated an adverse effect on strength, and the compressive strength decreased at a high ratio of H/S (20). When the H/S ratio increased from 12 to 20, the 28-day strength changed from 18.5 MPa to 23.3 MPa (M/S = 15), 19.2 MPa to 29.3 MPa (M/S = 18), 22.3 MPa to 39.6 MPa (M/S = 20), and 20.5 MPa to 31.4 MPa (M/S = 25), and this strength at 3 days rose from 7.0 MPa to 12.3 MPa (M/S = 15), 7.6 MPa to 18.3 MPa (M/S = 18), 10.7 MPa to 22.1 MPa (M/S = 20), 9.0 MPa to 20.3 MPa (M/S = 25). For the 3-day and 28-day strength, an H/S of 18 can generate the largest strength value. This phenomenon is caused by the influence of water, the low H/S led to low content of water, and the ion migration speed became slow and the hydration reaction was limited, causing a relatively low content of hydrated phases; thus, the strength was low, but excess water in the system could’t be consumed completely, which generated capillary pores in matrix, leading to low strength. Therefore, a suitable H/S ratio is important for MOS cement, and an H/S of 18 is appropriated for the porous MOS cement preparation due to high strength at 3 days and 28 days. 

#### 3.1.3. Modifier

To further improve the hardened performance of MOS cement paste, Wang et al. and Qin et al. investigated the influence of various admixtures; their research studies indicated that citric acid (C_6_H_8_O_7_⋅H_2_O) and ethylene diamine tetraacetic acid can be used to promote the generation of 517 phase for enhancing the hardened performance. In this research, composite modifier was added to the MgO-MgSO_4_-H_2_O system to improve the matrix strength, as shown in Figure 5. When the dosage of the admixture rose from 0 to 0.26%, the 3-day compressive strength rose between 22.1 MPa and 53.7 MPa and then decreased to 47.5 MPa. This strength at 28 days rose from 39.6 MPa to 73.8 MPa and then reduced to 61.7 MPa. When the content of modifier is 0.16%, this composite admixture plays a significant influence and promotes the generation of the largest strength value. Therefore, 0.16% of admixture was added to the MgO-MgSO_4_-H_2_O system to obtain high-strength matrix.

### 3.2. Effect of Bentonite-to-Water Ratio on Performance of Porous MOS Cement Paste

As mentioned above, 0.16% of modifier and 20MgO-MgSO_4_·7H_2_O-18H_2_O were suitable for porous MOS cement matrix preparation. To prepare the porous cement paste, bentonite slurry used as a void-forming agent was introduced into matrix. Prior to this, the bentonite-to-water ratio of the void-forming agent should be identified by comparing their hardened performance. From Figure 6, the strength of samples fabricated by 40% and 60% of bentonite slurry with various bentonite-to-water ratios indicate that bentonite slurry with a bentonite-to-water ratio of 1:5 was suitable for porous cement paste preparation. The reason is that the 3-day and 28-day strength rapidly reduced from 13.2 MPa to 10.0 MPa and 19.4 MPa to 14.4 MPa, respectively, when the bentonite-to-water ratio changed from 1:2 to 1:5, but the dry bulk density also decreased, which is beneficial to obtain lightweight porous material. However, further changing this ratio to 1:15, the strength decreased but the dry bulk density changed slightly. Combined with the results of samples with 60% bentonite slurry (Figure 6), it can be seen that a similar tendency occurred under 3 days’ curing, and at 28 days, a bentonite-to-water ratio of 1:5 promoted the strength increase, and further change in bentonite-to-water ratio showed the same trend. When the content of bentonite slurry was 40%, a 1:2 ratio generated the highest strength, but also caused the highest dry bulk density. Although the strength was not the highest in porous cement paste with a ratio of 1:5, it possessed a relatively low dry bulk density. When both reasons are taken into account, 1:5 was the best choice for porous cement paste. For the density of samples, the change of bentonite to slurry from 1:2 to 1:15 led to more water introduction, but the water was not fully consumed by the hydration reaction, causing the formation of capillary pores and decreased dry bulk density. For strength, the decreasing dry bulk density was negative for the strength; thus, the strength reduced with the changing bentonite-to-water ratio. However, for high content of bentonite slurry (60%), there are enough nanolayers in the bentonite slurry for void refinement to enhance strength, but a high concentration of bentonite slurry is adverse for montmorillonite layer exfoliation to generate nanolayers. A low concentration of bentonite slurry increases the water content, which is adverse for strength. Combined with the results above, a bentonite-to-water ratio of 1:5 is suitable for balanced void generation and refinement, and shows the highest strength at 28 days. For 3 days, hydration products were not enough and the strength of matrix was relatively low for preventing the increasing voids from excess water introduction; thus, it showed a continuous reduction in strength with the decreasing concentration of bentonite slurry.

### 3.3. Pore Structure and Hardened Performance of Porous MOS Cement Paste

#### 3.3.1. Pore Structure

As shown above, the 0.16% admixture-modified 20MgO-MgSO_4_·7H_2_O-18H_2_O system is suitable for the preparation of porous MOS cement matrix, and bentonite slurry with a bentonite-to-water ratio of 1:5 was injected into MOS cement matrix to obtain porous cement paste. After 28 day’s curing, the pore structure of the samples was characterized by MIP. Figure 7 indicates that introducing bentonite slurry into the cement matrix caused the increasing porosity. When the volume replacement of the bentonite slurry rose between 0 and 60%, the porosity of MOS cement paste increased linearly from 21.1% to 58.1%. Different from traditional porous MOS cement pastes, the pore morphology of these samples indicates that there are no macroscopic visible pores, such as air voids (Figure 8). Furthermore, increased porosity caused by bentonite slurry led to a change in pore size, when bentonite slurry rose from 20% to 60%, SEM images showed that the pore size increased, but compared with blank (bentonite slurry 0%), the pore size of porous MOS cement paste became smaller. To further analyze this change, the pore size distribution (PSD) was obtained by MIP, as shown in Figure 9. With the increasing porosity caused by introducing bentonite slurry (34.8–58.1%), the PSD curve moved to the direction with a large pore size value, and the pore size of the samples rose significantly; the size corresponding to highest pore volume in this curve increased from 22.8 nm to 163.0 nm. However, although no bentonite slurry was added into the system, the blank sample possesses the largest pore size; the peak value associated with largest pore volume in the PSD curve is 180.0 nm. 

Pore generation is associated with a pore-forming agent. The existing state of the montmorillonite nanolayer in the system was investigated via XRD. Figure 10 shows that the sharp 001 peak associated with montmorillonite in bentonite disappeared in porous MOS cement paste, indicating that montmorillonite exists as single nanolayers, because montmorillonite platelets or layers can be exfoliated in water to form nanolayers. When the bentonite slurry was injected into the MOS cement matrix, these nanolayers were also brought into the cement matrix, thus the water in bentonite slurry cannot be consumed completely by hydration reaction, and excess water in the system formed a large number of big capillary pores. However, these nanolayers from bentonite slurry were added into capillary voids, refining the big pore space and contributing to nanopore space generation. For the blank cement matrix, no nanolayer was added into system, and the capillary space could not be refined and these pores were still big, as shown in Figure 9. In addition, the excess water from the increased bentonite slurry content formed more and larger spaces, but the nanolayers in bentonite slurry were limited, which could not refine all pores, finally leading to increased pore size when bentonite slurry dosage rose between 20% and 60%.

#### 3.3.2. Hardened Performance

Figure 11 shows the variation of dry bulk density of porous MOS cement paste. The dry bulk density reduced linearly between 1650 kg/m^3^ and 760 kg/m^3^; this can be attributed to the rise in porosity due to the introduction of bentonite slurry, as mentioned above. Figure 12 presents the compressive strength of porous MOS cement paste: the 3-day strength and 28-day strength reduced from 58.1 MPa to 5.0 MPa and 68.8 MPa to 7.8 MPa with increasing content of bentonite slurry, respectively. Considering the relationship between dry bulk density and bentonite slurry content, the relationship between strength and density can be obtained (Figure 13). When the dry bulk density rose from 760 kg/m^3^ to 1650 kg/m^3^, the 3-day curing compressive strength increased from 5.0 MPa to 58.1 MPa, and the strength during 28 days’ curing rose between 7.8 MPa and 69.8 MPa. The variation of strength could be due to the decrease in dry bulk density caused by the high content of bentonite paste. Specifically, the decreased density caused the reduction in strength. Moreover, because of the generation of more hydration products, the compressive strength increased with the prolonging of curing age.

Figure 14 presents the change in the thermal conductivity value of porous MOS cement paste when bentonite slurry increased between 0% and 60%. Specifically, with increasing content from 0 to 60%, the thermal conductivity of porous MOS cement paste dropped from 0.85 W/(m·K) to 0.25 W/(m·K), indicating that the thermal insulation property of the MOS-based material was enhanced. The enhanced thermal insulation performance is attributed to the decreased dry bulk density, because it can generate a large quantity of pores, which benefits the reduction in thermal conductivity. Combined with data in Figure 11 and Figure 14, the relationship can be drawn, as shown in Figure 15, that the increased bentonite contributed to the decreased dry bulk density, with the reduction in dry bulk density from 1650 kg/m^3^ to 760 kg/m^3^; and the related thermal conductivity value of porous MOS cement paste linearly dropped from 0.85 W/(m·K) to 0.25 W/(m·K). When the dry bulk density was 760 kg/m^3^, the thermal conductivity value was 0.25 W/(m·K), meaning that MOS cement-based materials present an insulation performance, which can be used in the building energy-saving field. 

## 4. Conclusions

Porous magnesium oxysulfate (MOS) cement pastes can be successfully fabricated by introducing presaturated bentonite to modified MOS cement paste. The main results are summarized as follows:The results indicated that the 20MgO-MgSO_4_·7H_2_O-18H_2_O system modified by citric acid (C_6_H_8_O_7_⋅H_2_O) and ethylene diamine tetraacetic acid was suitable to be used as matrix to fabricate porous MOS cement paste.Bentonite slurry with a bentonite-to-water ratio of 1:5 was suitable for porous cement paste preparation, due to the decreased dry bulk density and relatively high compressive strength of cement matrix.Significant refinement of pores can be achieved by adding bentonite slurry, contributing to the generation of nanosized pores in MOS cement pastes. When bentonite slurry content in MOS cement paste rose from 0 to 60%, the peak that occurred at the pore size distribution curve of the MOS cement-based materials decreased from 180.0 nm to 22.8 nm and then moved to 163.0 nm, and the porosity linearly increased from 21.1% to 58.1%.Porous MOS cement paste with a thermal conductivity of 0.25–0.85 W/(m·K), compressive strength of 7.8–69.8 MPa, and dry bulk density of 760–1650 kg/m^3^ was obtained by adding 1:5 of bentonite slurry into modified cement matrix, showing great potential as a green thermal insulation material in building energy saving and CO_2_ emission reduction.

## Figures and Tables

**Figure 1 materials-15-06872-f001:**
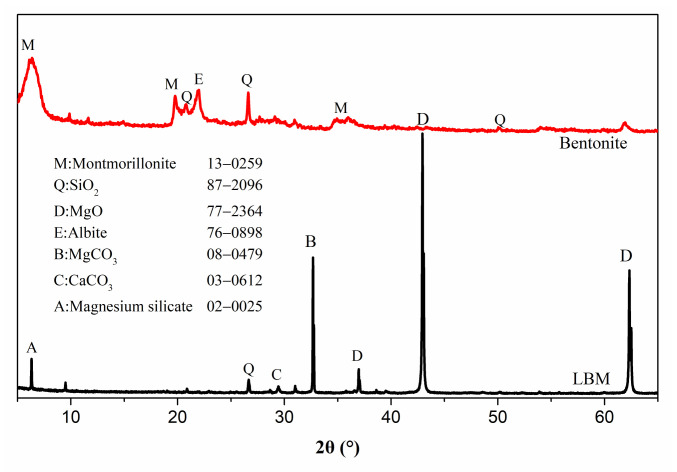
XRD curves of bentonite and LBM.

**Figure 2 materials-15-06872-f002:**
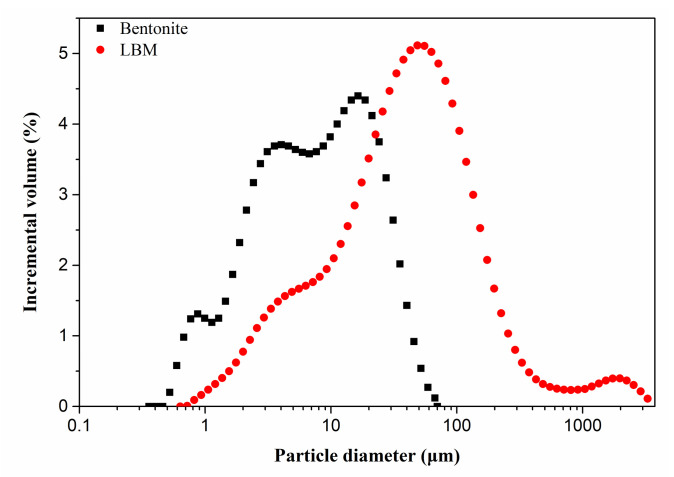
Particle size distribution of LBM and bentonite.

**Figure 3 materials-15-06872-f003:**
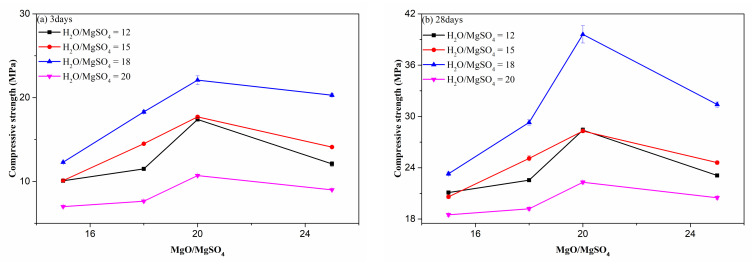
Effect of M/S on the compressive strength of cement paste.

**Figure 4 materials-15-06872-f004:**
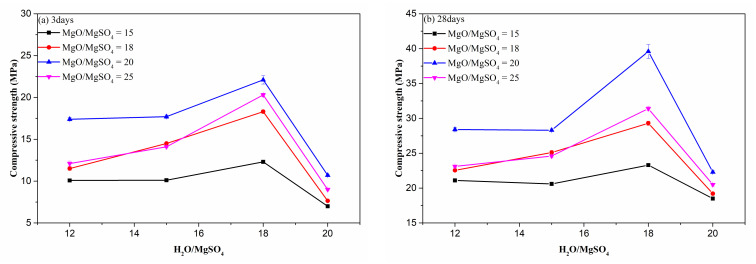
Influence of H/S on the compressive strength of MOS cement matrix.

**Figure 5 materials-15-06872-f005:**
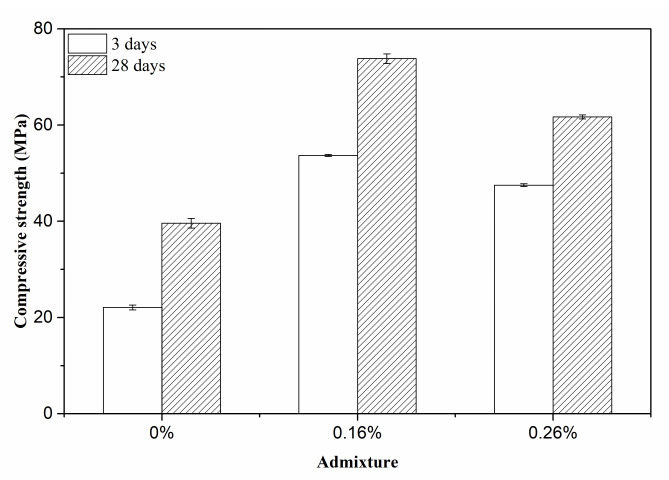
Influence of modifier on the compressive strength of MOS cement matrix.

**Figure 6 materials-15-06872-f006:**
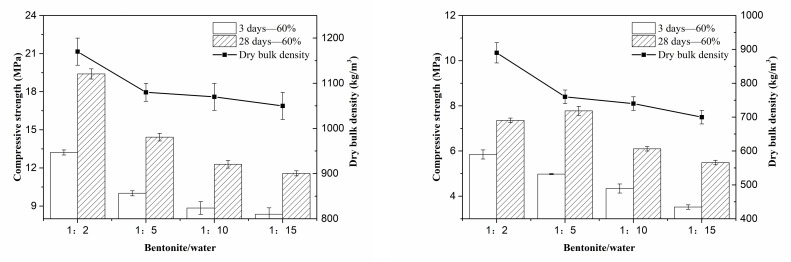
Influence of bentonite-to-water ratio on the compressive strength of porous MOS cement paste.

**Figure 7 materials-15-06872-f007:**
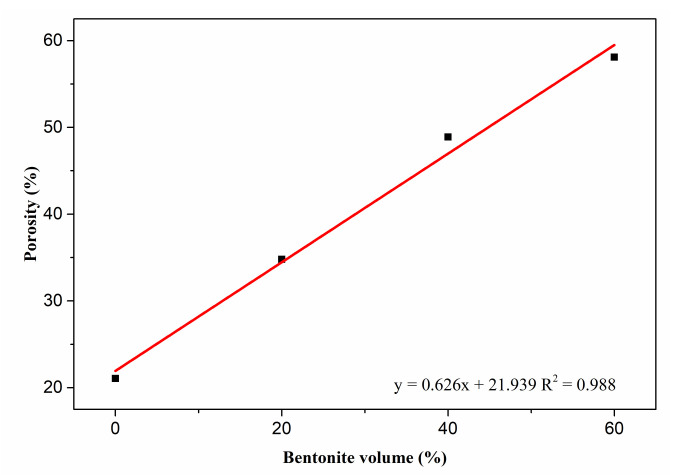
Porosity of porous MOS cement paste.

**Figure 8 materials-15-06872-f008:**
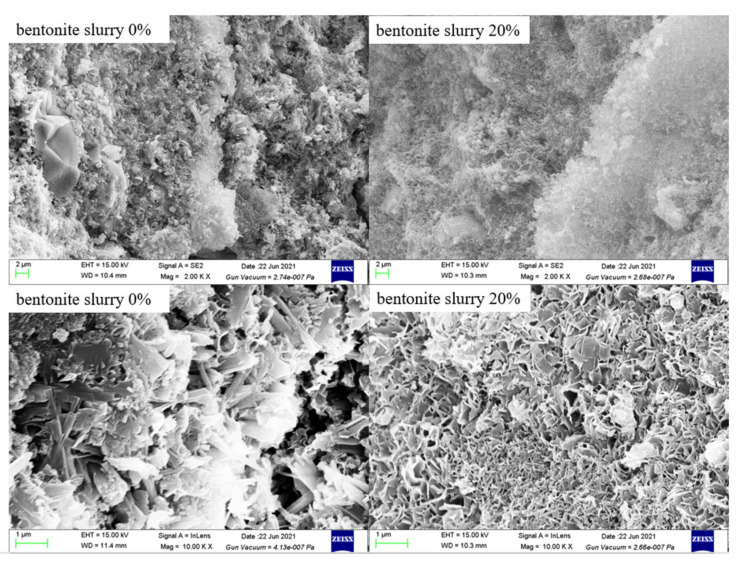
SEM images of porous samples and matrix.

**Figure 9 materials-15-06872-f009:**
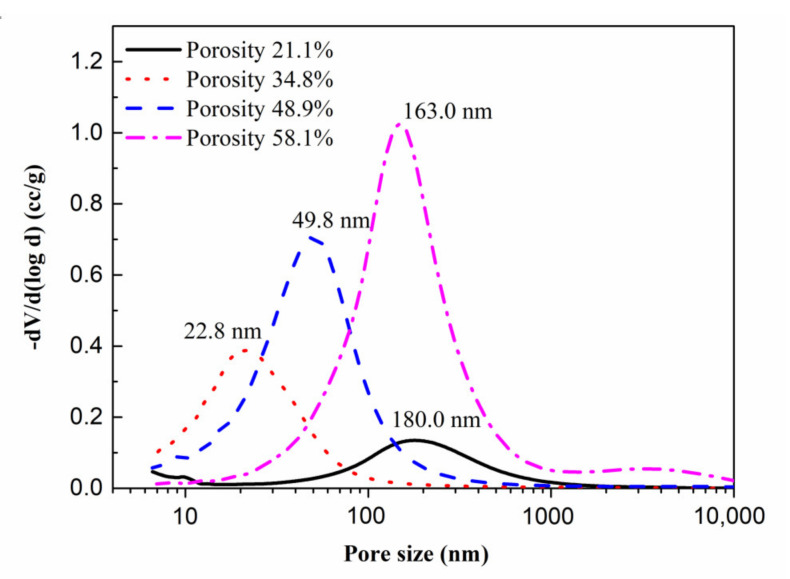
Pore size distribution of porous MOS cement paste.

**Figure 10 materials-15-06872-f010:**
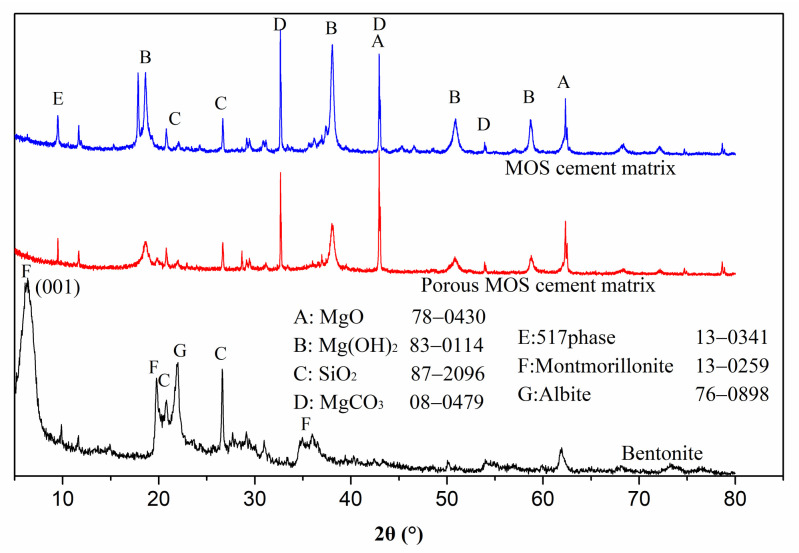
XRD patterns of porous MOS cement paste and cement matrix.

**Figure 11 materials-15-06872-f011:**
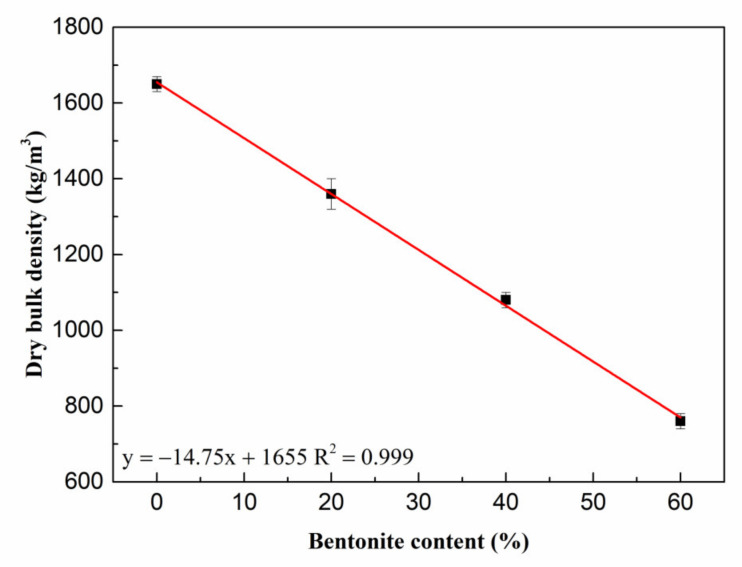
Dry bulk density of porous samples.

**Figure 12 materials-15-06872-f012:**
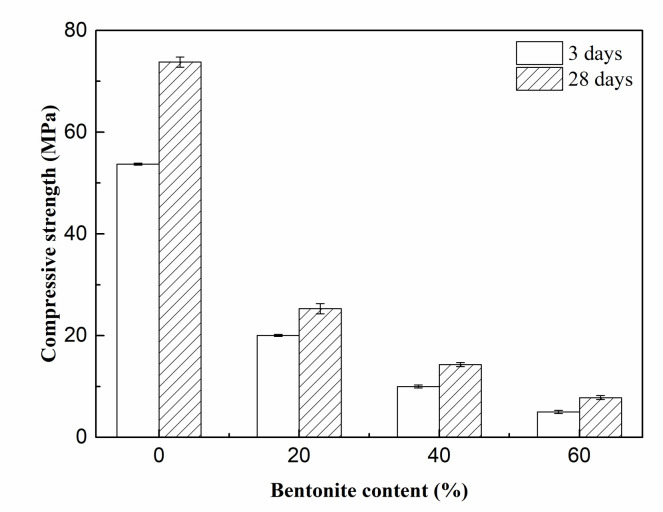
Compressive strength of porous samples.

**Figure 13 materials-15-06872-f013:**
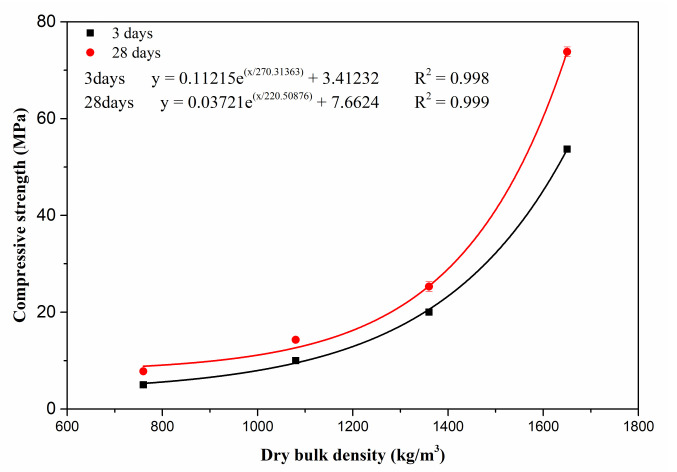
Relationship between dry bulk density and compressive strength.

**Figure 14 materials-15-06872-f014:**
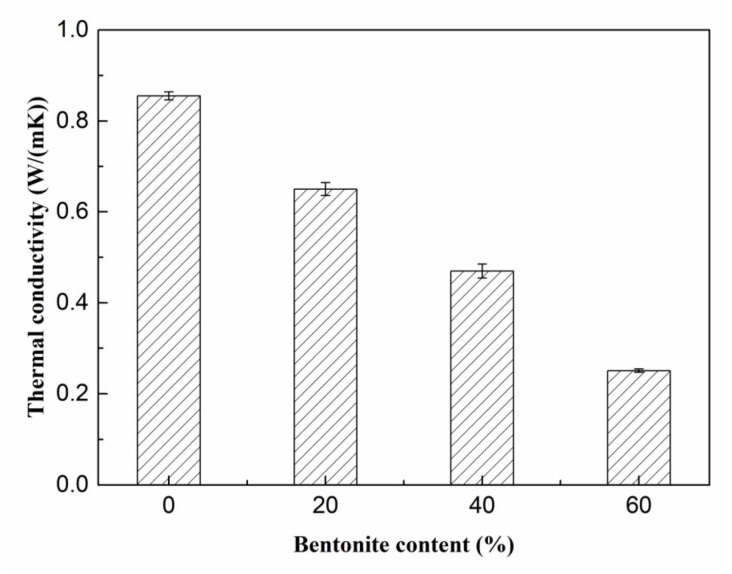
Thermal conductivity of porous samples.

**Figure 15 materials-15-06872-f015:**
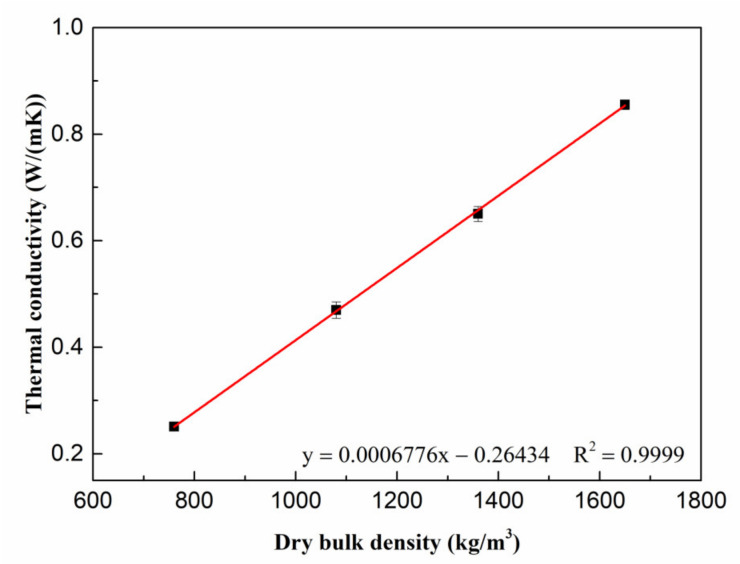
Relationship between thermal conductivity and dry bulk density.

**Table 1 materials-15-06872-t001:** Chemical composition of LBM and bentonite.

Compositions (%)	SiO_2_	Al_2_O_3_	Fe_2_O_3_	CaO	MgO	SO_3_	K_2_O
LBM	6.81	1.07	0.64	3.94	87.01	0.29	0.04
Bentonite	75.75	15.30	1.77	3.02	1.78	1.35	0.60

**Table 2 materials-15-06872-t002:** Mix parameters of porous MOS cement paste and MOS cement paste.

MOS Cement Paste			Bentonite to Cement Ratio	MOS Cement Paste (m^3^)	Bentonite Gel (m^3^)
MgO to MgSO_4_ Mole Ratio	H_2_O to MgSO_4_ Mole Ratio	Admixture			
15	12	-	-	-	-
18	12	-	-	-	-
20	12	-	-	-	-
25	12	-	-	-	-
15	15	-	-	-	-
18	15	-	-	-	-
20	15	-	-	-	-
25	15	-	-	-	-
15	18	-	-	-	-
18	18	-	-	-	-
20	18	-	-	-	-
25	18	-	-	-	-
15	20	-	-	-	-
18	20	-	-	-	-
20	20	-	-	-	-
25	20	-	-	-	-
20	18	0.16%	-	-	-
20	18	0.26%	-	-	-
20	18	0.36%	-	-	-
20	18	0.16%	1:2	0.6	0.40
20	18	0.16%	1:5	0.6	0.40
20	18	0.16%	1:10	0.6	0.40
20	18	0.16%	1:15	0.6	0.40
20	18	0.16%	1:2	0.40	0.60
20	18	0.16%	1:5	0.40	0.60
20	18	0.16%	1:10	0.40	0.60
20	18	0.16%	1:15	0.40	0.60
20	18	0.16%	1:5	0.8	0.20

## Data Availability

The raw/processed data required to reproduce these findings cannot be shared at this time as the data also form part of an ongoing study.

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
