# Peer review of "Preparation and Hardened Performance of Bentonite-Induced Porous Magnesium Oxysulfate Cement Paste"

_materials, 2022, doi:10.3390/ma15196872_

Round 1

Reviewer 1 Report

This a relatively extensive study reporting on the optimisation of composite materials based on magnesium oxide-sulfide and bentonite with respect to their applications mainly for thermal insulation (as a replacement of Portland cement). The presented results, their interpretations and the drawn conclusions look sound. So in my view, the paper can be published in Materials in the current version, provided the English will be substantially improved. I have only some technical notes:

- I am not sure about the units in the last two columns in Table 2. Are these realy m^3?

- The information provided in Fig. 3 is essentially the same as that in Fig. 2.  Moreover, a detailed description of of the figure contents is given in the text. So Fig. 3 should be removed. 

- It is not clear why the bentonite to water ratio 1:5 was adopted as the optimal one even for 40% of bentonite slurry, where the compressive strength is the highest for 1:2 ratio.

- Fig. 10 - please indicate the 001 peak of monmorrilonite. It would be nice to perform a Rietveld refinement to get the quantitative phase composition.

- I would prefer to use a term dry bulk density or dry apparent density instead of dry density only (as the pores are included in the reference volume)

Reviewer 2 Report

This is an interesting study, investigating bentonite-induced porous magnesium oxysulfate cement pastes.  However, before the publication some parts of the manuscript should be improved. Please find some comments and suggestions bellow.

Line 100-108: If the chemical composition, mineral phases and particle size distribution analysis were performed by the authors (in frame of this study, not provided from the literature), then Figure1, Figure 2 and Table 1 (and their description) are showing results and should be moved into Results section.

Line 136: I think “X-ray fluorescence spectrometer (XRF)…” would be more correct, because you are talking about device, not measurement technique.

Line 138-140: Why “afterward”? This part is not clear, please modify. Which type of alcohol was used? Why leaching mode (isn`t it dispersion?)? Please check and correct.

Line 142: In order to avoid repeating the same expression “Until its mass didn`t change” could be replaced by “to the constant mass”.

Line 142-147:

XRD: X-ray diffraction of clay minerals requires a special protocol of sample preparation (separation of particles <2µm by centrifugation in aqueous suspension, heating…). Please describe the protocol of sample preparation. Additionally, the used software and crystallographic data base should be given. Please list also the ICSD codes for all determined mineral phases.

SEM/EDX: Please describe the conditions for SEM measurements (like vacuum mode, kV, working distance…) and sample preparation (polished/unpolished, if any coating...). Please add clear scale bars on Figure 8. In “2.2 test procedures” it is mentioned that elements were detected by EDX. In the results I didn`t find EXD results. If EDX was performed, it should be measured on polished samples. Please check and correct in the manuscript.
